# RTFormer: Efficient Design for Real-Time Semantic Segmentation with Transformer

**Jian Wang**[1]* **Chenhui Gou**[2]* **Qiman Wu**[1]* **Haocheng Feng**[1]
**Junyu Han**[1] **Errui Ding**[1] **Jingdong Wang**[1]†
[1]Baidu VIS    [2]Australian National University(ANU)
{wangjian33, wuqiman, fenghaocheng, hanjunyu, dingerrui, wangjingdong}@baidu.com
u7194588@anu.edu.au

## Abstract

Recently, transformer-based networks have shown impressive results in semantic segmentation. Yet for real-time semantic segmentation, pure CNN-based approaches still dominate in this field, due to the time-consuming computation mechanism of transformer. We propose RTFormer, an efficient dual-resolution transformer for real-time semantic segmenation, which achieves better trade-off between performance and efficiency than CNN-based models. To achieve high inference efficiency on GPU-like devices, our RTFormer leverages GPU-Friendly Attention with linear complexity and discards the multi-head mechanism. Besides, we find that cross-resolution attention is more efficient to gather global context information for high-resolution branch by spreading the high level knowledge learned from low-resolution branch. Extensive experiments on mainstream benchmarks demonstrate the effectiveness of our proposed RTFormer, it achieves state-of-the-art on Cityscapes, CamVid and COCOStuff, and shows promising results on ADE20K. Code is available at PaddleSeg[24]: https://github.com/PaddlePaddle/PaddleSeg.

## 1 Introduction

Semantic segmentation is a fundamental computer vision task which usually serves as critical perception module in autonomous driving, mobile applications, robot sensing and so on. In pace with the development of these applications, the demand of executing semantic segmentation in real-time grows stronger increasingly. Existing real-time segmentation methods mainly focus on exploiting CNN architectures, including designing high-efficiency backbones and decoders[48, 42, 41, 17, 21, 13, 6, 28] by handcraft and exploring neural architecture search methods to find better trade-off between accuracy and efficiency[47, 23, 22, 9]. And significant improvement has been achieved by these great works so far.

More recently, vision transformers have been drawn lots of attention for their strong visual recognition capability[12, 31, 25, 35, 36]. And inherited from them,

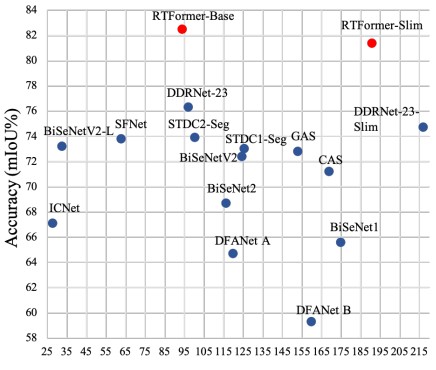

Figure 1: **Accuracy(mIoU%) vs. Inference Speed(FPS) on CamVid[4] test set.** Our methods are presented in red dots while other methods are presented in blue dots.

---

*Equal Contribution.
†Corresponding author.

36th Conference on Neural Information Processing Systems (NeurIPS 2022).

a series of transformer-based architectures like[51, 38, 44, 30] are proposed and show very promising performance on general semantic segmentation task. Comparing with CNN-based networks, the main distinction of these transformer-based architectures is the heavy usage of self-attention, and self-attention is good at capturing long-range context information which is essential in semantic segmentation. So, we consider that attention structure should also be effective in real-time semantic segmentation task.

But up to now, only a few works[38] have explored the application of attention in this field, and the state-of-the-arts are still dominated by CNN-based architectures. We suppose the main obstacles of applying attention in real-time setting might come from the following two aspects. One is that the computation properties of most existing types of attention are not quite inference friendly for GPU-liked devices, such as the quadratic complexity and the multi-head mechanism. The quadratic complexity introduces large computation burden when processing high resolution features, especially in dense prediction tasks like semantic segmentation. Although several works such as[35, 38] shrink down the size of keys and values, the property of quadratic complexity remains. While the multi-head mechanism splits the matrix multiplication into multiple groups, which makes the attention operation to be time consuming on GPU-like devices, analogue to the situation that executing group convolution. The other is that conducting attention only on high resolution feature map itself like[38, 44] may not be the most effective way for capturing long-range context with high level semantic information, as a single feature vector from high resolution feature map has limited receptive field.

We propose a novel transformer block, named RTFormer block, as shown in Figure 2, which aims to achieve better trade-off between performance and efficiency on GPU-like devices with transformer. For the low-resolution branch, a newly proposed GPU-Friendly Attention, derived from external attention[15], is adopted. It inherits the linear complexity property from external attention, and alleviates the weakness of multi-head mechanism for GPU-like devices by means of discarding the channel split within matrix multiplication operations. Instead, it enlarges the number of external parameters and splits the second normalization within double-norm operation proposed by external attention into multiple groups. This enables GPU-Friendly Attention to be able to maintain the superiority of multi-head mechanism to some extent. For the high-resolution branch, we adopt cross-resolution attention instead of only conducting attention within high-resolution feature itself. Besides, unlike the parallel formulation of multi-resolution fusion from[32, 17, 44], we arrange the two resolution branches into a stepped layout. Therefore, the high-resolution branch can be enhanced more effectively by the assistant of the high level global context information learned from low-resolution branch. Based on the proposed RTFormer block, we construct a new real-time semantic segmentation network, named RTFormer. In order to learn enough local context, we still use convolution blocks at the earlier stages and place RTFormer block at the last two stages. By taking extensive experiments, we find RTFormer can make use of global context more effectively and achieve better trade-off than previous works. Figure 1 shows the comparison between RTFormer with other methods on CamVid. Finally, we summarize the contribution of RTFormer as following three aspects:

- A novel RTFormer block is proposed, which achieves better trade-off between performance and efficiency on GPU-like devices for semantic segmentation task.

- A new network architecture RTFormer is proposed, which can make full use of global context for improving semantic segmentation by utilizing attention deeply without lost of efficiency.

- RTFormer achieves state-of-the-art on Cityscapes, CamVid and COCOStuff, and show promising performance on ADE20K. In addition, it provides a new perspective for practice on real-time semantic segmentation task.

## 2   Related Work

**Generic Semantic segmentation.** Traditional segmentation methods utilized the hand-crafted features to solve the pixel-level label assigning problems,e.g., threshold selection[29],the super-pixel[1] and the graph algorithm[3]. With the success of deep learning, a series of methods[7, 2, 50] based on FCN (fully convolutional neural network) [26] achieve superior performance on various benchmarks. These methods improved FCN from different aspects. The Deeplabv3 [7] and the PSPNet[50] enlarge the receptive field and fused different level features by introducing the atrous spatial pyramid pooling

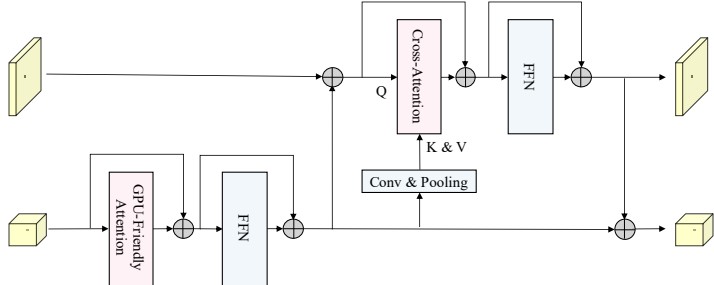

Figure 2: Illustration of RTFormer block. For low resolution, GPU-Friendly Attention is applied. And for high resolution, we use Cross-resolution Attention which draws K and V from low resolution branch. Besides, we make up FFN with two $3 \times 3$ convolution layers.

module and the pyramid pooling module. The SegNet[2] recovers the high-resolution map through the encoder-decoder structure. HRNet [32] introduces a multi-resolution architecture which maintains high-resolution feature maps all through the network. OCRNet [43] enhances the feature outputted from backbone by querying global context.

**Real-time Semantic segmentation.** To solve the real-time segmentation problem, various methods[48, 42, 41, 17, 13, 9] have been proposed. ICNet[48] solves this problem by using a well-designed multi-resolution image cascade network. FasterSeg[9] utilizes neural architecture search (NAS) to reach the goal of balancing high accuarcy and low latency. BiSeNetV1[42] and BiSeNetV2 [41] adopt a two-stream paths network and a feature fusion module to achieve a well balance between speed and segmentation performance. STDC[13] rethinks and improves BiSeNet by proposing a single-steam structure with detail guidance module. DDRNets[17] achieves better performance by designing a two deep branch network with multiple bilateral fusions and a Deep Aggregation Pyramid Pooling Module.

**Attention mechanism.** The attention mechanism has been vigorous developed in computer vision field [37, 45, 18, 19, 14, 39]. SE block proposed by [19] applies the attention function to channels and improves the representation capability of the network. [18] uses an attention-module to model object relation and help the objection detection. [37] presents a non-local operation which can capture the long-range dependencies and shows promising results on video classification task. [39] uses attention in point cloud recognition task. Self-attention is a special case of the attention mechanism that has been widely used in recent years[45, 14, 33]. However, the quadratic complexity limits its usage. Some works[15, 34, 40] reform self-attention to achieve linear complexity. But they are still not friendly for inference on GPU. Inspired by external attention [15], we developed a GPU-Friendly attention module that has low latency and high performance on GPU-like devices.

**Transformer in Semantic segmentation.** Very recently, transformer shows promising performance in semantic segmentation. DPT [30] applies transformer as encoder to improve the performance of dense prediction task. SETR [51] proposes a sequence-to sequence method and achieves impressing result. SETR uses pretrained VIT [46] as its backbone and has no downsampling in spatial resolution. However, it is difficult to use it for real-time segmentation task due to its heavy backbone and very high resolution. SegFormer [38] increases efficiency by introducing a hierarchical transformer encoder and a lightweight all MLP decoder. Compared with SETR, SegFormer has both higher efficiency and higher performance. However, the efficiency of SegFormer is still relatively low compared to some state-of-the-art CNN based real-time segmentation model. By introducing our RTFormer block, our method can take advantage of the attention mechanism while achieving the real-time speed.

## 3  Methodology

In this section, we elaborate the details of our proposed approach. We first describe the RTFormer block, then we present how to construct RTFormer based on RTFormer block.

### 3.1 RTFormer block

RTFormer block is dual-resolution module which inherits the multi-resolution fusion paradigm from[32, 17, 44]. In contrast to the previous works, RTFormer block is comprised of two types of attention along with their feed forward network, and arranged as stepped layout, as shown in Figure 2. In the low-resolution branch, we use a GPU-Friendly Attention to capture high level global context. While in the high-resolution branch, we introduce a cross-resolution attention to broadcast the high level global context learned from low-resolution branch to each high-resolution pixel, and the stepped layout is served to feed more representative feature from the low-resolution branch into the cross-resolution attention.

**GPU-Friendly Attention.** Comparing the different existing types of attention, we find that external attention[15](EA) can be a potential choice for being executed on GPU-like devices due to its gratifying property of linear complexity, and our GPU-Friendly Attention(GFA) is derived from it. Thus, before detailing GFA, we review EA first. Let $X \in \mathbb{R}^{N \times d}$ denotes an input feature, where $N$ is the number of elements(or pixels in images) and $d$ is the feature dimension, then the formulation of EA can be expressed as:

$$EA(X, K, V) = DN(X \cdot K^T) \cdot V \qquad (1)$$

where $K, V \in \mathbb{R}^{M \times d}$ are learnable parameters, $M$ is the parameter dimension, $DN$ is the Double Normalization operation proposed by[15]. And the multi-head version of EA can be expressed as:

$$MHEA(X) = Concat(h_1, h_2, ..., h_H)$$
$$h_i = EA(X_i, K', V'), \quad i \in [1, H] \qquad (2)$$

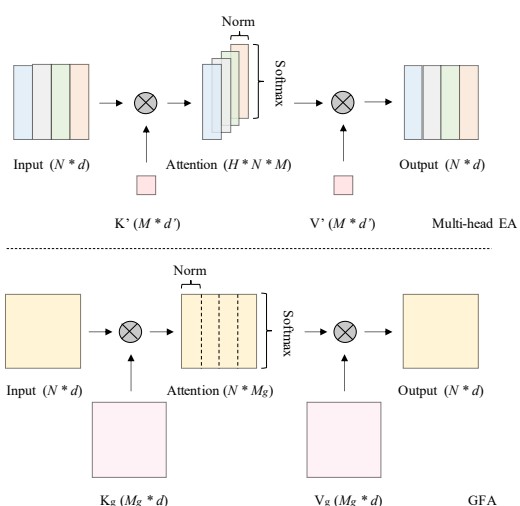

Figure 3: Comparison between Multi-Head External Attention and GPU-Friendly Attention. Multi-head external attention splits the matrix multiplication into several groups while our GPU-Friendly Attention makes matrix multiplication to be integrated which is more friendly for GPU-like devices.

where $K', V' \in \mathbb{R}^{M \times d'}$, $d' = d/H$ and $H$ is the number of heads, while $X_i$ is the $i$th head of $X$. As shown in upper part of Figure 3, the multi-head mechanism generates $H$ attention maps for improving upon the capacity of EA, and this makes the matrix multiplication be splitted into several groups, which is similar as group convolution. Although EA uses shared $K'$ and $V'$ for different heads, which can speed up the calculation a lot, the splitted matrix multiplication remains.

To avoid the latency reduction on GPU-like devices due to the multi-head mechanism, we propose a simple and effective GPU-Friendly Attention. It evolves from the basic external attention expressed by Equation 1, which can be formulated as:

$$GFA(X, K_g, V_g) = GDN(X \cdot K_g^T) \cdot V_g \qquad (3)$$

where $K_g, V_g \in \mathbb{R}^{M_g \times d}$, $M_g = M \times H$ and $GDN$ denotes Grouped Double Normalization, which splits the second normalization of the original double normalization into $H$ groups, as shown in the left lower part of Figure 3. From Equation 3 we can find that GFA has two main improvements. On the one hand, it makes the matrix multiplication to be integrated, which is quite friendly for GPU-like devices. Benefit from this, we can enlarge the size of external parameters from $(M, d')$ to $(M_g, d)$. Therefore, more parameters can be tuned for improving the performance. On the other hand, it maintains the superiority of multi-head mechanism to some extent by taking advantage of the grouped double normalization. For intuitive comprehension, it can be regarded that GFA also generates $H$ different attention maps for capturing different relations between tokens, but more feature elements are involved for computing similarity and all the attention maps contribute to the final output.

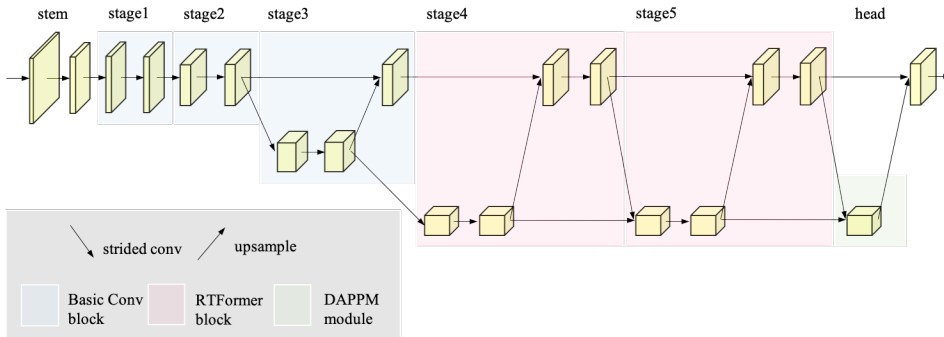

Figure 4: Illustrating the RTFormer architecture. We place RTFormer block at the last two stages which indicated by pink block and use convolution blocks at the earlier stages which indicated by blue block. Besides, we add a DAPPM module for segmentation head, drawing on the successful experience from[17].

**Cross-resolution Attention.** Multi-resolution fusion has been proven to be effective for dense prediction task. And for the design of multi-resolution architecture, we can intuitively apply the GFA in different resolution branches independently, and exchange features after the convolution module or attention module being executed like[32, 44]. But in high-resolution branch, pixels focus on local information more than high level global context. Thus, we suppose that directly conducting attention on high-resolution feature map for learning global context is not effective enough. To obtain the global context more effectively, we propose a cross-resolution attention, which aims to make full use of the high level semantic information learned from low-resolution branch. As exhibited in Figure 2, unlike GFA, cross-resolution attention is adopted in high-resolution branch for gathering global context. And the calculation of this cross-resolution attention is expressed as:

$$CA(X_h, K_c, V_c) = Softmax(\frac{X_h \cdot K_c^T}{\sqrt{d_h}}) \cdot V_c$$

$$K_c, V_c = \phi(X_c) \quad X_c = \theta(X_l) \tag{4}$$

where $X_h$, $X_l$ denote the feature maps on high-resolution branch and low-resolution branch respectively, $\phi$ is a set of matrix operations including splitting, permutation and reshaping, $d_h$ means the feature dimension of high-resolution branch. It is worth to explain that, the feature map $X_c$, denoted as cross-feature in the following text, is computed from $X_l$ by function $\theta$ which is composed of pooling and convolution layers. And the spatial size of $X_c$ indicates the number of tokens generated from the $X_l$. Experimentally, we only adopt softmax upon the last axis of attention map for normalization, as a single softmax performs better than double normalization when the key and value are not external parameters. Specially, for fast inference on GPU-like devices, multi-head mechanism is also discarded here.

**Feed Forward Network.** In the previous transformer-based segmentation methods like[38, 44], the Feed Forward Network(FFN) is typically consist of two MLP layers and a depth-wise $3 \times 3$ convolution layer, where the depth-wise $3 \times 3$ layer works for supplementing position encoding or enhancing locality. Besides, the two MLP layers expand the hidden dimension to be two or four times of the input dimension. This type of FFN can achieve better performance with relative less parameters. But in the scenario where latency on GPU-like devices should be considered, the typical structure of FFN is not very efficient. In order to balance the performance and efficiency, we adopt two $3 \times 3$ convolution layers without dimension expansion in the FFN of RTFormer block. And it shows even better result than the typical FFN configuration.

## 3.2 RTFormer

Figure 4 illustrates the overall architecture of RTFormer.

**Backbone Architecture.** For extracting enough local information which is needed by high-resolution feature map, we combines convolution layers with our proposed RTFormer block to construct

Table 1: **Detailed configurations of architecture variants of RTFormer.**

| Models | #Channels | #Blocks | Spatial size of cross-feature |
|---|---|---|---|
| RTFormer-Slim | $[32, 64, 64/128, 64/256, 64/256]$ | $[2, 2, 1/2, 1, 1]$ | $8 \times 8$ |
| RTFormer-Base | $[64, 128, 128/256, 128/512, 128/512]$ | $[2, 2, 1/2, 1, 1]$ | $12 \times 12$ |

RTFormer. Concretely, we let RTFormer start from a stem block consist of two $3 \times 3$ convolution layers and make up the first two stages with several successive basic residual blocks[16]. Then, from stage3, we use dual-resolution modules which enable feature exchange between high-resolution and low-resolution branches, inspired by[17]. And for the high-resolution branches of the last three stages, the feature strides keep as $8$ unchanged, while for the low-resolution branches, the feature strides are $16$, $32$, $32$ respectively. Specially, we arrange the dual-resolution module into stepped layout for boosting the semantic representation of high-resolution feature with the help of the output of low-resolution branch. Most importantly, we construct the stage4 and stage5 with our proposed RTFormer block which is illustrated in Figure 2 for efficient global context modeling, while the stage3 is still composed by basic residual blocks.

**Segmentation Head.** For the segmentation head of RTFormer, we add a DAPPM module after low-resolution output feature, drawing on the successful experience from[17]. And after fusing the output of DAPPM with high-resolution feature, we obtain the output feature map with stride=$8$. Finally, this output feature is passed into a pixel-level classification head for predicting dense semantic labels. And the classification head is consist of a $3 \times 3$ convolution layer and a $1 \times 1$ convolution layer, with the hidden feature dimension being same with input feature dimension.

**Instantiation.** We instantiate the architecture of RTFormer with RTFormer-Slim and RTFormer-Base, and the detailed configurations are recorded in Table 1. For the number of channels and number of blocks, each array contains $5$ elements, which are corresponding to the $5$ stages respectively. Especially, the elements with two numbers are corresponding to the dual-resolution stages. For instance, $64/128$ means the number of channels is $64$ for high-resolution branch and $128$ for low-resolution branch. While $1/2$ means the number of basic convolution blocks is $1$ for high-resolution branch and $2$ for low-resolution branch. It is worth to be noted that, the last two elements in block number array denote the number of RTFormer blocks, and they are both $1$ for RTFormer-Slim and RTFormer-Base. The spatial sizes of cross-feature are set as $64(8 \times 8)$ and $144(12 \times 12)$ for RTFormer-Slim and RTFormer-Base respectively.

# 4 Experiments

In this section, we valid RTFormer on Cityscapes[10], Camvid[4], ADE20K[52] and COCOStuff[5]. We first introduce the datasets with their training details. Then, we compare RTFormer with state-of-the-art real-time methods on Cityscapes and CamVid. Besides, more experiments on ADE20K[52] and COCOStuff[5] are summarised to further prove the generality of our method. Finally, ablation studies of different design modules within RTFormer block on ADE20K[52] are provided.

## 4.1 Implementation Details

Before finetuning on semantic segmentation, all models are pretrained on ImageNet[11]. And the training details for ImageNet[11] will be provided in the supplementary material. We apply mIoU and FPS as the metrics for performance and efficiency respectively, and the FPS is measured on RTX 2080Ti without tensorrt acceleration by default.

**Cityscapes.** Cityscapes[10] is a widely-used urban street scene parsing dataset, which contains 19 classes used for semantic segmentation task. And it has 2975, 500 and 1525 fine annotated images for training, validation, and testing respectively. We train all models using the AdamW optimizer with the initial learning rate 0.0004 and the weight decay of 0.0125. We adopt the poly learning policy with the power of 0.9 to drop the learning rate and implement the data augmentation method including random cropping into $512 \times 1024$, random scaling in the range of 0.5 to 2.0, and random horizontal flipping. All models are trained with 484 epochs (about 120K iterations), a batch size of 12, and syncBN on four V100 GPUs. For a fair comparison with other algorithms, online hard example mining(OHEM) is not used.

Table 2: **Comparisons with other state-of-the-art real-time methods on Cityscapes and CamVid.** Performances are measured with a single crop of $1024 \times 2048$, $720 \times 960$ for Cityscapes and CamVid respectively. #Params refers to the number of parameters. FPS is calculateted under the same input scale as performance measuring. In this table, * means we retrain this method follows its original training setting, and $\dotplus$ means we measure the FPS on single RTX 2080Ti GPU.

| Method | Encoder | #Params↓ | GPU | Cityscapes | | CamVid | |
|---|---|---|---|---|---|---|---|
| | | | | FPS↑ | val mIoU(%)↑ | FPS↑ | test mIoU(%)↑ |
| ICNet [49] | - | - | TitanX M | 30.3 | 67.7 | 27.8 | 67.1 |
| DFANet A [20] | Xception A | 7.8M | TitanX | 100.0 | - | 120.0 | 64.7 |
| DFANet B [20] | Xception B | 4.8M | TitanX | 120.0 | - | 160.0 | 59.3 |
| CAS [47] | - | - | TitanX | 108.0 | 71.6 | 169.0 | 71.2 |
| GAS [23] | - | - | TitanX | 108.4 | 72.4 | 153.1 | 72.8 |
| DF1-Seg-d8 [22] | DF1 | - | GTX 1080Ti | 136.9 | 72.4 | - | - |
| DF1-Seg [22] | DF1 | - | GTX 1080Ti | 106.4 | 74.1 | - | - |
| DF2-Seg1 [22] | DF2 | - | GTX 1080Ti | 67.2 | 75.9 | - | - |
| DF2-Seg2 [22] | DF2 | - | GTX 1080Ti | 56.3 | 76.9 | - | - |
| BiSeNet1 [42] | Xception39 | 5.8M | GTX 1080Ti | 105.8 | 69.0 | 175.0 | 65.6 |
| BiSeNet2 [42] | ResNet18 | 49.0M | GTX 1080Ti | 65.5 | 74.8 | 116.3 | 68.7 |
| BiSeNetV2[41] | - | - | GTX 1080Ti | 156.0 | 73.4 | 124.5 | 72.4 |
| BiSeNetV2-L [41] | - | - | GTX 1080Ti | 47.3 | 75.8 | 32.7 | 73.2 |
| SFNet [21] | ResNet18 | 12.9M | RTX 2080Ti | - | - | 62.9$\dotplus$ | 73.8 |
| FasterSeg [9] | - | 4.4M | RTX 2080Ti | 136.2$\dotplus$ | 73.1 | - | 71.1 |
| STDC1-Seg75[13] | STDC1 | 14.2M | RTX 2080Ti | 74.6$\dotplus$ | 74.5 | - | - |
| STDC2-Seg75[13] | STDC2 | 22.2M | RTX 2080Ti | 73.5$\dotplus$ | 77.0 | - | - |
| STDC1-Seg[13] | STDC1 | 14.2M | RTX 2080Ti | - | - | 125.6$\dotplus$ | 73.0 |
| STDC2-Seg[13] | STDC2 | 22.2M | RTX 2080Ti | - | - | 100.5$\dotplus$ | 73.9 |
| DDRNet-23-Slim [17] | - | 5.7M | RTX 2080Ti | 101.0$\dotplus$ | 76.1 | 217.0$\dotplus$ | 74.7 |
| DDRNet-23 [17] | - | 20.1M | RTX 2080Ti | 38.3$\dotplus$ | 78.9* | 97.1$\dotplus$ | 76.3 |
| **RTFormer-Slim**(Ours) | - | 4.8M | RTX 2080Ti | 110.0$\dotplus$ | 76.3 | 190.7$\dotplus$ | 81.4 |
| **RTFormer-Base**(Ours) | - | 16.8M | RTX 2080Ti | 39.1$\dotplus$ | 79.3 | 94.0$\dotplus$ | 82.5 |

**CamVid.** CamVid[4] contains 701 densely annotated frames and the resolution of each frame is $720 \times 960$. These frames are divided into 367 training images, 101 validation images, and 233 testing images. CamVid[4] have 32 categories which has the subset of 11 classes are used for segmentation experiments. We merge the training set and validation set for training and evaluate our models on the testing set. We set the initial learning rate to 0.001 and the weight decay to 0.05. The power of poly learning policy is set to 1.0. We train all models for 968 epochs. Data augmentation includes color jitter, random horizontal flipping, random cropping into $720 \times 960$ and random scaling of [288, 1152]. Unlike previous methods[13], we do not pretrain our model on Cityscapes[10]. All other training details are the same as for Cityscapes[10].

**ADE20K.** ADE20K[52] is a scene parsing dataset covering 150 fine-grained semantic concepts, which split 20K, 2K, and 3K images for training, validation, and testing, respectively. Our models are trained with a batch size of 16 for 160k iterations. And we set the initial learning rate to 0.0001 and the weight decay to 0.05, and the other training settings are identical to those for Cityscapes[10].

**COCOStuff.** COCOStuff[5] is a dense annotated dataset derived from COCO. It contains 10K images (9K for training and 1K for testing) with respect to 182 categories, including 91 thing and 91 stuff classes. And 11 of the thing classes have no annotations. We train RTFormer 110 epochs on COCOStuff with AdamW optimizer, and the initial learning rate and weight decay are set as 0.0001 and 0.05 respectively. In the training phase, we first resize the short side of image to 640 and randomly crop $640 \times 640$ patch for augmentation. While in the testing phase, we resize all images into $640 \times 640$. Other training settings are identical to Cityscapes.

## 4.2 Comparison with State-of-the-arts

In this part, we compare our RTFormer with state-of-the-art methods on Cityscapes[10] and CamVid[4]. Table 2 shows our results including parameters, FPS and mIoU for Cityscapes[10] and CamVid[4].

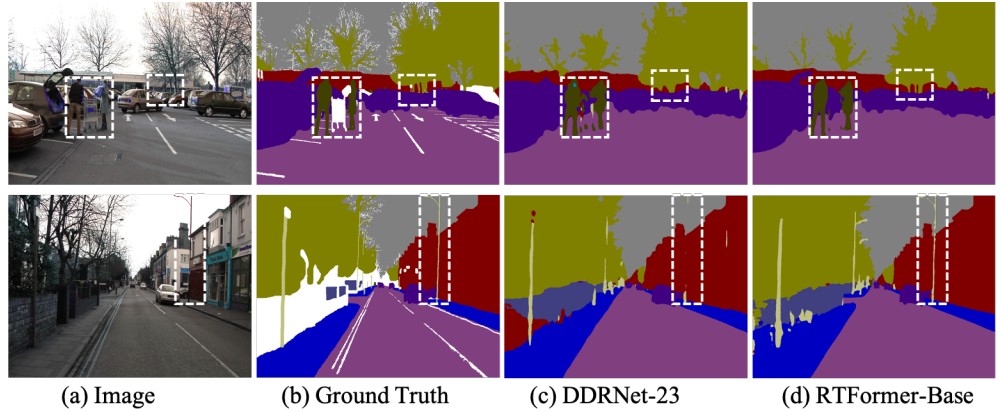

| (a) Image | (b) Ground Truth | (c) DDRNet-23 | (d) RTFormer-Base |

Figure 5: **Qualitative results on CamVid[4] testing set.** Compared to DDRNet-23[14], RTFormer predicts masks with finer details and reduces long-range errors as highlighted in white.

Table 3: **Comparisons with other state-of-the-art real-time methods on ADE20K.** The #Params, FLOPs and FPS are measured at resolution $512 \times 2048$. #Params refers to the number of parameters. In this table, * means that we retrain this model by ourself on ADE20K[52]. $\dotplus$ means we measure the FPS using single RTX 2080Ti GPU. Method without $\dotplus$ is using its reported #Params, FLOPs, FPS and mIoU.

| Method | Encoder | #Params↓ | FLOPs↓ | FPS↑ | val mIoU(%)↑ |
|---|---|---|---|---|---|
| FCN [27] | MobileNetV2 | 9.8M | 39.0G | 64.4 | 19.7 |
| PSPNet[50] | MobileNetV2 | 13.7M | 52.9G | 57.7 | 29.6 |
| DeepLabV3+ [8] | MobileNetV2 | 15.4M | 69.4G | 43.1 | 34.0 |
| SegFormer [38] | MiT-B0 | 3.8M | 8.4G | 50.5 | 37.4 |
| DDRNet-23-Slim [17] | - | 5.6M | 18.2G | 189.1$\dotplus$ | 33.3* |
| DDRNet-23[17] | - | 20.1M | 71.6G | 71.2$\dotplus$ | 38.8* |
| **RTFormer-Slim**(Ours) | - | 4.8M | 17.5G | 187.9 | 36.7 |
| **RTFormer-Base**(Ours) | - | 16.8M | 67.4G | 71.4 | 42.1 |

**Results.** On Cityscapes[10], our RTFormer owns the best speed-accuracy trade-off among all other real-time methods. For example, our RTFormer-Slim achieves 76.3% mIoU at 110.0 FPS which is faster and provides better mIoU compared to STDC2-Seg75[13] and DDRNet-23-Slim[14]. Besides, our RTFormer-Base achieves 39.1 FPS and 79.3% mIoU which establishes new state-of-the-art result. Further more, using only ImageNet[11] pre-training, our method achieves 82.5% mIoU at 94.0 FPS on CamVid[4], significantly outperforms all other real-time methods including STDC2-Seg[13] which uses additional Cityscapes[10] pre-training. Moreover, Our RTFormer-Slim yields 81.4 mIoU at 190.7 FPS with only 4.8M, which is faster and better than other models like STDC2-Seg[13] at 125.6FPS and DDRNet-23[17] at 97.1FPS. Figure 5 shows the qualitative results on CamVid[4] testing set, where RTFormer-base provides better detail than DDRNet-23[17], especially for the Column Pole class , which requires more global context. In summary, these results demonstrate the superiority of RTFormer in real-time semantic segmentation in terms of accuracy, latency, and model size.

### 4.3 Generalization Capability

To further prove the effectiveness of our RTFormer on more generalized scene, we show additional results on ADE20K[52] and COCOStuff[5].

**Results.** Table 3 presents our result on ADE20K[52]. Our RTFormer-Base archieves the superior mIoU of 42.1% and with 71.4FPS, which outperforms all other methods. For instances, in contrast to DDRNet-23-Slim[17], RTFormer-Slim achieves better mIoU 36.7% and maintains nearly the same speed. Figure 6 shows qualitative results on ADE20K validation set. Compared with DDRNet-23[17],

Table 4: **Comparisons with other state-of-the-art real-time methods on COCOStuff.** The #Params, FLOPs and FPS are measured at resolution $640 \times 640$.

| Method | GPU | #Params↓ | FLOPs↓ | FPS↑ | test mIoU(%)↑ |
|---|---|---|---|---|---|
| PSPNet50[50] | - | - | - | 6.6 | 32.6 |
| ICNet[49] | TitanX M | - | - | 35.7 | 29.1 |
| BiSeNetV2[41] | GTX 1080Ti | - | - | 87.9 | 25.2 |
| BiSeNetV2-L[41] | GTX 1080Ti | - | - | 42.5 | 28.7 |
| DDRNet-23 | RTX 2080Ti | 20.1M | 28.1G | 146.1 | 32.1 |
| **RTFormer-Base** | RTX 2080Ti | 16.8M | 26.6G | 143.3 | 35.3 |

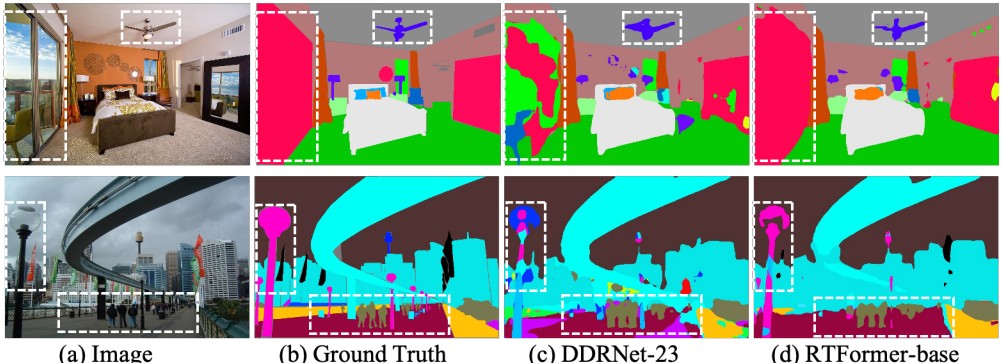

(a) Image      (b) Ground Truth      (c) DDRNet-23      (d) RTFormer-base

Figure 6: **Qualitative results on ADE20K[52] validation set.** As shown, RTFormer is good at focusing on global context.

our RTFormer shows better details and context information. In summary, these results demonstrate that RTFormer also shows very promising performance on real-time semantic segmentation in generalized scene. While on COCOStuff, as shown in Table 4, our RTFormer-Base achieves 35.3 mIoU at 143.3 FPS, which outperforms the DDRNet-23 about 3% with a comparable inference speed and sets a new state-of-the-art.

### 4.4 Ablation study on ADE20K

**Training Setup.** We provide ablation results with RTFormer-Slim. To make quick evaluations, we train RTFormer-slim from scratch with the initial learning rate being set to 0.001, and the other training settings are same with experiments on ADE20K[52] above. More experimental details and analyses are elaborated in the supplementary material.

**Comparison on different types of attention.** To verify the effectiveness of our proposed attentions, we replace the attentions used in RTFormer block with different types and combinations. As shown in Table 5a, we give the results of different combinations of multi-head self attention, multi-head external attention, GPU-Friendly attention and cross-resolution attention. For instance, "GFA+CA" means using GFA in low-resolution branch and CA in high-resolution branch. In addition, we adjust the hyper parameter $M$ in multi-head external attention by $M = d \times r$, where $r$ is a reduction ratio. We can find that GPU-Friendly attention outperforms all settings of multi-head external attention and is faster than the best one when $M = d$, and meanwhile, GPU-Friendly attention is much more efficient than multi-head self attention with comparable performance. That indicates GPU-Friendly attention achieves better trade-off between performance and efficiency than both multi-head self attention and multi-head external attention on GPU-like devices. When we introduce cross-resolution attention, the performance improves further, while the FPS only drops less than 2.

**Comparison on different types of FFN.** Table 5b illustrates the results of typical FFN which is consist of two MLP layers and a $3 \times 3$ depth-wise convolution layer and our proposed FFN containing two $3 \times 3$ convolution layers. It is shown that our proposed FFN outperforms typical FFN not only

Table 5: **Ablation studies on different types of attention, FFN and different settings of hyper parameters.**

(a) Comparison on different types of attention. SA, EA, GFA, CA denote Self Attention, External Attention, GPU-Friendly Attention and Cross-resolution Attention respectively.

| Attention | FPS↑ | mIoU(%)↑ |
|---|---|---|
| SA+SA | 97.4 | 32.7 |
| EA+EA (r=1) | 180.8 | 32.2 |
| EA+EA (r=0.125) | 196.9 | 31.9 |
| EA+EA (r=0.25) | 189.6 | 32.0 |
| GFA+GFA | 189.8 | 32.8 |
| GFA+CA | 187.9 | 33.0 |

(b) Comparison of different types of FFN. The typical FFN is composed of two MLP layers and a $3 \times 3$ depthwise convolution layer, and our FFN design is two $3 \times 3$ convolution layers.

| Method | FPS↑ | mIoU(%)↑ |
|---|---|---|
| Typical FFN | 178.5 | 32.15 |
| Our FFN | 187.9 | 33.0 |

(c) Comparison of different number of groups in Grouped Double Normalization.

| # of Groups | FPS↑ | mIoU(%)↑ |
|---|---|---|
| 1, 1 | 189.8 | 32.2 |
| 4, 1 | 189.8 | 32.3 |
| 8, 2 | 189.8 | 32.8 |

(d) Comparison of different spatial size of the cross-feature in Cross-resolution Attention.

| # of Parameters | FPS↑ | mIoU(%)↑ |
|---|---|---|
| $6 \times 6$ | 191.6 | 32.85 |
| $8 \times 8$ | 189.8 | 33.00 |
| $12 \times 12$ | 175.6 | 32.94 |

on mIoU but also on FPS. That indicates that our proposed FFN is more suitable in the scenario when latency on GPU-like devices should be considered.

**Influence of the number of groups within grouped double normalization.** We study the influence of the number of group in grouped double normalization under the setting of using GPU-Friendly Attention for both branches. And Table 5c shows the results of different configurations. For example, "8+2" means using $8$ groups in low-resolution branch and $2$ groups in high-resolution. Specially, when the number of groups is set to $1$, grouped double normalization degrades to the original double normalization. Here, the best mIoU is achieved when the numbers of groups are $8$ and $2$, which illustrates that the grouped double normalization performs better than the original double normalization. And it is worth to be noted that, changing the number of groups in grouped double normalization does not affect the inference efficiency, which makes GPU-Friendly attention being able to keep high FPS when the number of groups is large.

**Influence of the spatial size of cross-feature in Cross-resolution Attention.** We also investigate the spatial size of cross-feature in cross-resolution attention, including applying $6 \times 6$, $8 \times 8$, and $12 \times 12$. As presented in Table 5d, $8 \times 8$ spatial size of cross-feature for RTFormer-Slim is the best according to the trade-off between FPS and mIoU. To some extent, it indicates that the spatial size of cross-feature which is close to the dimension of high-resolution feature is appropriate, as the high-resolution feature dimension of RTFormer-Slim is $64$ which equals to $8 \times 8$.

## 5 Conclusion

In this paper, we present RTFormer which can efficiently capture the global context to improve the real-time semantic segmentation performance. Extensive experiments demonstrate that our method not only achieves new state-of-the-art results on common datasets for real-time segmentation but also shows superior performance on challenging dataset for general semantic segmentation. Due to the efficiency of RTFormer, we hope our method can encourage new design of real-time semantic segmentation with transformer. One limitation is that while our RTFormer-Slim only has $4.8$M parameters, more parameter efficiency may be needed in a chip of edge device. We leave it for future work.

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
