# A    ImageNet Pre-training

RTFormer is consist of several convolution blocks and RT-Former blocks, and RTFormer block contains different types of attention. Thus, we pre-train RTFormer on ImageNet-1K(1) mainly following the settings of training transformer network(8), and the detail configuration is provided in Table 1.

Table 2 shows the performance of RTFormer on ImageNet classification. Both RTFormer-Slim and RTFormer-Base outperform the corresponding DDRNet variants. In addition, RTFormer-Base achieves the best performance among the existing backbones adopted in real-time semantic segmentation task.

# B    More Experiments

In this section, we extend the ablation study about different types of attention. Firstly, we supplement experimental details about different types of attention, meanwhile, we introduce more variants of attention for analysis. Then, we analyse the results of different types of attention in detail.

Table 1: Training settings on ImageNet classification.

| config | value |
|---|---|
| optimizer | AdamW |
| base learning rate | 0.0005 |
| weight decay | 0.04 |
| optimizer momentum | $\beta_1, \beta_2 = 0.9, 0.999$ |
| batch size | 1024 |
| learning rate schedule | cosine decay |
| minimum learning rate | 5e-6 |
| warmup epochs | 5 |
| warmup learning rate | 5e-7 |
| training epochs | 300 |
| augmentation | RandAug(9, 0.5) |
| color jitter | 0.4 |
| mixup | 0.2 |
| cutmix | 1.0 |
| random erasing | 0.25 |
| drop path | 0.0 |

## B.1    Experimental Details.

The self-attention used for comparison is following (12). In contrast to the traditional self-attention, this type of self-attention shrinks the spatial size of key and value as $\frac{1}{\sigma}$ of the input feature, which can reduce the computation cost caused by the large input resolution. We set $\sigma = 4$ for the self-attention in high-resolution branch, while $\sigma = 1$ for low-resolution branch, following the settings for feature maps with stride=8 and stride=32 in(12).

For both multi-head self-attention and multi-head external attention, which are denoted as SA and EA in Table 3, we set the number of heads as 2 and 8 for high-resolution and low-resolution branches respectively. Similarly, for the GPU-Friendly attention, we set the number of groups as 2 and 8 separately for high-resolution and low-resolution branches. For the case of GFA+CA, the number of groups of the GPU-Friendly attention in low-resolution is still set as 8, while the cross-resolution attention has no multi-head calculation.

Especially for multi-head external attention, we give several results with different hyper parameters for comprehensive comparison. The first three results of multi-head external attention are with $r = [0.125, 0.25, 1]$ respectively. When $r = 0.25$, the parameter dimension of multi-head external attention $M$ in low-resolution branch is 64, which is identical to the setting in(3). And the other two results are used for showing more variations of the trade-off between performance and inference speed. In addition, an extra result with $r = 1, C = 36$ is given, where $C$ is the number of base feature dimension in network($C = 32$ for RTFormer-Slim by default). For GPU-Friendly attention, we set $M_g = d$ constantly.

Further more, we also compare with the attentions proposed in Linformer (11) and Nyströmformer(13). For linformer attention, we directly give a result without hyper parameter modification. While for nyströmformer attention, we give two results denoted as NA(32) and NA(64), which differs in the number of landmark points.

## B.2    Analyses.

As illustrated in Table 3, we can find that multi-head self-attention achieves 32.7 mIoU, which performs better than multi-head external attentions with different settings of $r$. But, the inference speed of multi-head self-attention is not competitive, which is mainly caused by the quadratic complexity and multi-head mechanism.

Multi-head external attention can achieve a good inference speed, which is benefit from its linear complexity and the design of sharing external parameter for multiple heads. Associated with the

Table 2: **Classification accuracy on the ImageNet validation set.** Performances are measured with a single $224 \times 224$ crop. "#Params" refers to the number of parameters. "FLOPs" is calculated under the input scale of $224 \times 224$.

| Method | #Params↓ | FLOPs↓ | Top-1 Acc. ↑ |
|---|---|---|---|
| ResNet-18(4) | 11.2M | 1.8G | 69.0 |
| RestNet-50(4) | 23.5M | 3.7G | 75.3 |
| DF1(7) | 8.0M | 0.7G | 69.8 |
| DF2(7) | 17.5M | 1.7G | 73.9 |
| MobileNetV2(9) | 3.4M | 0.3G | 72.0 |
| MobileNetV3(6) | 5.4M | 0.2G | 75.2 |
| Efficient-Net-B0(10) | 5.3M | 0.4G | 76.3 |
| STDC1(2) | 8.4M | 0.8G | 73.9 |
| STDC2(2) | 12.5M | 1.4G | 76.4 |
| DDRNet-23-slim(5) | 7.6M | 1.0G | 70.2 |
| DDRNet-23(5) | 28.2M | 3.9G | 75.9 |
| **RTFormer-Slim** | 5.3M | 0.8G | 72.3 |
| **RTFormer-Base** | 20.5M | 3.0G | 77.4 |

Table 3: **Comparison among different types of attention on ADE20K.** SA, EA, GFA, CA, LA, NA denote multi-head self-attention, multi-head external attention, GPU-Friendly attention, cross-resolution attention, linformer attention and nyströmformer attention respectively. For example, GFA+CA means adopting GFA in low-resolution branch and CA in high-resolution branch. $r$ is a ratio for adjusting the parameter dimension $M$ in multi-head external attention. $C$ is the number of base feature dimension in network ($C = 32$ by default). NA(32), NA(64) denote the nyströmformer attention with 32 and 64 landmark points respectively.

| Attention | GPU | FPS↑ | val mIoU(%)↑ |
|---|---|---|---|
| SA+SA | RTX 2080Ti | 97.4 | 32.7 |
| EA+EA (r=0.125) | RTX 2080Ti | 196.9 | 31.9 |
| EA+EA (r=0.25) | RTX 2080Ti | 189.6 | 32.0 |
| EA+EA (r=1) | RTX 2080Ti | 180.8 | 32.2 |
| EA+EA(r=1,C=36) | RTX 2080Ti | 134.8 | 32.8 |
| LA+LA | RTX 2080Ti | 167.6 | 32.4 |
| NA(32)+NA(32) | RTX 2080Ti | 77.6 | 32.9 |
| NA(64)+NA(64) | RTX 2080Ti | 72.2 | 33.0 |
| GFA+GFA | RTX 2080Ti | 189.8 | 32.8 |
| GFA+CA | RTX 2080Ti | 187.9 | 33.0 |

above two properties, multi-head external attention adopts a low parameter dimension $M(\ll d)$, which reduces the total computation cost further. However, the performance of multi-head external attention is suboptimal, as the network capacity is limited by those designs. Yet, the multi-head mechanism still remains, which is not friendly for running on GPU-like devices and leads to a relative worse efficiency than single head situation. As a example, when we let $M$ to be equal to $d$, the performance is still worse than multi-head self-attention, and the inference speed drops about 10FPS than $M = 0.25d$.

The linformer attention achieves linear complexity by projecting the keys and values to a space where token length is fixed. But it is still built upon multi-head mechanism. The nyströmformer attention repurposes the nyström method for approximating self-attention computation, and it achieves linear complexity by adopting landmark points to reconstruct the softmax matrix. However, it splits the original softmax matrix computation into several parts which causes the suboptimal inference efficiency on GPU-like devices. Besides of the splitting operation, nyströmformer also has the problem brought by the vanilla multi-head mechanism.

While, GPU-Friendly attention, which is derived from multi-head external attention, can achieve both relative good performance and inference speed. It is because that, GPU-Friendly attention discards the multi-head mechanism and makes the matrix multiplication to be integrated and friendly for GPU calculation. Meanwhile, the grouped double normalization in GFA helps to maintain the capacity for learning diverse information which can be regarded as an extension of multi-head mechanism. Therefore, the external parameters can be enlarged for increasing the network capacity without great loss of inference speed.

Further more, when the basic feature dimension $C$ is enlarged from 32 to 36 for EA+EA($r = 1$), the mIoU increases to 32.8, while the FPS drops from 180.8 to 134.8. From this result, we can conclude that the network equipped with GFA+GFA is faster than EA+EA about 41% when they achieve the same performance, and this improvement is considerable.

Finally, the combination of GPU-Friendly attention and cross-resolution attention improves the performance further, and it outperforms other types and combinations of attentions in both accuracy and efficiency, which validates the effectiveness of our proposed attentions.