# OpenReview forum: "RTFormer: Efficient Design for Real-Time Semantic Segmentation with Transformer"
_NeurIPS.cc/2022/Conference — NeurIPS 2022 Accept_

### Official Review · Reviewer_jocB · 2022-07-09

**Rating:** 5
**Confidence:** 5
**Soundness:** 2 fair
**Presentation:** 2 fair
**Contribution:** 2 fair

**Summary:**

This paper proposes RTFormer, an efficient transformer for real-time semantic segmentation, which achieves better trade-off between performance and efficiency than CNN-based models. To achieve high inference efficiency on GPU-like devices, the RTFormer leverages GPU-Friendly Attention with linear complexity and discards the multi-head mechanism. Besides, the cross-resolution attention is more efficient to gather global context information for high-resolution branch by spreading the high level knowledge learned from low-resolution branch. Extensive experiments on mainstream benchmarks demonstrate the effectiveness of the proposed RTFormer, it achieves state-of-the-art on Cityscapes and CamVid, and shows promising results on ADE20K

**Questions:**

The questions are as follows:
1. is it possible to generalise the proposed network to other vision tasks, like image classification, object detection, etc.
2. It seems the attention used is just MHA or self-attention, this is not a new technique. So the main contribution is limited, right?


**Ethics Review Area:**

["I don’t know"]

**Limitations:**

yes

**Strengths And Weaknesses:**

Strengths of this paper are as follows :
1. A novel RTFormer block is proposed, which achieves better trade-off between performance and efficiency on GPU-like devices for semantic segmentation task.
2. A new network architecture RTFormer is proposed, which can make full use of global context for improving semantic segmentation by utilizing attention deeply without lost of efficiency.
3. RTFormer achieves state-of-the-art on Cityscapes and CamVid, and show promising performance on ADE20K. In addition, it provides a new perspective for practice on real-time semantic segmentation task.

Weakness of this paper are as follows:
1. The proposed cross-resolution attention is just the variant of self-attention, which is widely used in network design. This paper is incremental compared with previous work DDRNet. The novelty of this paper is limited.
2. The performance  improvement on Cityscape dataset is limited. As shown in Table 1, the mIoU score and FPS improvement are both limited, not obvious enough.
3. The experiments on semantic segmentation is thorough enough and can be used to support the proposed method. But my concern is that the method is simple and not novel enough. In addition, why not apply this method to other vision tasks, like classification, object detection.

---

> ### Author Response · Authors · 2022-08-02
> **Reply to Reviewer jocB**
>
> ### Weakness 1: cross-resolution attention is just the variant of self-attention, and this paper is incremental compared with previous work DDRNet.
> > Cross-resolution attention is a technology used in RTFormer block for improving performance, which helps RTFormer to achieve better trade-off. And the proposed RTFormer is a new network, in which, the GPU-Friendly attention and RTFormer block designed for GPU characteristic are never discussed before, and the main novelty can be refered in the response of common question 1 and 2.
> ### Weakness 2: The performance improvement on Cityscape dataset is limited.
> > In this paper, Tabel 1 shows RTFormer’s parameters, FPS and mIoU over Cityscape dataset. As we can see, RTFormer-slim can achieve 0.2% mIoU than DDRNet-23-Slim with 19.3% less parameters and 1.1 multiple FPS. RTFormer-base has the same phenomenon. Transformer-based networks always have relatively small gain on Cityscape dataset. As we all know, SegFormer159
> MiTB0 only achieves 76.2% mIoU which lower than RTFormer-slim. Moreover, RTFormer can achieve constant improvements over CamVid, ADE20k and COCOStuff datasets, which proves that RTFormer has better generalization in terms of data.
> ### Weakness 3: Why not apply this method to other vision tasks, like classification, object detection.
> > Refer to Q.1 part.
> ### Question 1: Why not apply this method to other vision tasks, like classification, object detection.
> > Due to the time limitation, we did not explore the generalization of GFA on classification and detection tasks in this work, but it is planned in our future work.
> ### Question 2: the attention used is just MHA or self-attention, this is not a new technique.
> >  The MHA or self-attention has been widely used or developed by a some previous works such as External attention, Linformer and Nyströmformer, they reduce the attention computation cost from different perspective, however, their method may not suitable for real time application on GPU-like devices. For example, although Nyströmformer has linear computation complexity, our method is faster about 160% than Nyströmformer while keep the same performance. Compared with thse attention methods, GFA is more GPU-frendly and can acheive better speed-accuracy trade-off.

---

### Official Review · Reviewer_qoXt · 2022-07-10

**Rating:** 5
**Confidence:** 5
**Soundness:** 3 good
**Presentation:** 3 good
**Contribution:** 3 good

**Summary:**

This paper proposes RTFormer for real-time semantic segmentation.  The RTFormer leverages GPU-Friendly Attention with linear complexity and discards the multi-head mechanism. The authors demonstrate the efficacy of their method on several benchmarks.

**Questions:**

Please refer to “Paper Weaknesses”.

**Ethics Review Area:**

["I don’t know"]

**Limitations:**

Please refer to “Paper Weaknesses”.

**Strengths And Weaknesses:**

Strengths:
1. The proposed method achieves good performances on the benchmarks.
2. This paper is well organized and clearly described.
3. Efficient segmentation is a valuable problem.

Weaknesses:
1. The method proposed in the paper is a hybrid of various existing methods such as linear-complexity self-attention, HRNet, and CNN and transformer hybrid model. Therefore, the novelty is weakened by previous works.
2. This paper does not say whether to use tensorrt to accelerate the model. So I don't know if the comparison is fair.
3. In terms of performance and model size, there is no significant advantage between this method and the compared methods.

---

> ### Author Response · Authors · 2022-08-02
> **Reply to Reviewer qoXt**
>
> Dear Reviewer qoXt,
> Thank you for the detailed review. We will address your comments below.
> ### Weakness 1: the paper is a hybrid of various existing methods.
> > Refer to response of common question 1 and 2.
> ### Weakness 2: whether using tensorrt.
> > For fair comparison, we do not use tensorrt for acceleration in all of the experiments, and the compared methods which are re
> evaluated on RTX2080Ti are also tested without tensorrt.
> ### Weakness 3: there is no significant advantage in terms of performance and model size.
> > For model size, RTFormer is smaller that the DDRNet counterpart by about 16\%, which is not very margin. For the performance, only the improvement on cityscapes is relative small, and this situation is also appeared in other transformer-based network like SegFormer MiT-B0, which only achieves 76.2% mIoU with 15.2FPS. While for other benchmarks like ADE20K, COCOStuff, RTFormer achieves considerable improvements, which verifies its generalization ability. Espectially, on CamVid, RTFormer significantly outperforms previous real-time methods and sets new SOTA. And we argue that, it is not quite easy to show constant improvements on at least four mainstream benchmarks with the model size compressed.

---

> > ### Comment · Reviewer_qoXt · 2022-08-09
> > **Response**
> >
> > Thanks to the author's reply, my concerns were resolved. So I decided to raise my score.

---

> > > ### Author Response · Authors · 2022-08-09
> > > **Reply to Reviewer qoXt**
> > >
> > > Thanks for your reply and detailed review.
> > > Glad to know we have solved your concerns.

---

### Official Review · Reviewer_HQEz · 2022-07-11

**Rating:** 5
**Confidence:** 4
**Soundness:** 3 good
**Presentation:** 3 good
**Contribution:** 2 fair

**Summary:**

The manuscript presents an efficient model for semantic segmentation. The main contribution corresponds to GPU friendly attention layer which improves the efficiency by using keys and values as learnable parameters. Dimensionality of keys and values is a hyperparameter that is much less than N (HxW). Furthermore, the MLP from the standard transformer is replaced with plain convolutions. The resulting module is somewhat similar to the classic self-attention layer from pre-transformer era. Finally, some further performance improvement is obtained through cross-resolution attention. Experiments address Cityscapes and ADE20k.


**Questions:**

It appears that it would be more appropriate to call this model as a hybrid convolutional/attention model than a transformer.

It would be beneficial to show experiments on some embedded platform.

Please clarify whether you use V100 16GB or V100 32 GB.

**Limitations:**

It appears that large memory footprint precludes training on single GPU systems.

**Strengths And Weaknesses:**

Strengths
- a resonable hybrid model with convolutions on higher-resolution representations and self-attention on lower-resolution representations
- state-of-the-art ratio between performance and computational complexity

Weaknesses
- incremental contribution; GPU friendly attention appears quite related to previous work [13], as well as to Linformer and Nystromformer (see below).
- hybrid convolutional-transformer models have been proposed before (eg DPT hybrid, see below)
- the proposed improvements perform only slightly better than baselines in Fig.3
- missing configuration in Fig.3a: EA + CA
- large training footprint: it appears that only 3 crops 512x1024 can fit into a V100
- incomplete related work in the field of efficient models for semantic segmentation (eg HardNet, SwiftNet, see below)

Missing related work:
- René Ranftl, Alexey Bochkovskiy, Vladlen Koltun. Vision Transformers for Dense Prediction. ICCV 2021: 12159-12168
- Yunyang Xiong, Zhanpeng Zeng, Rudrasis Chakraborty, Mingxing Tan, Glenn Fung, Yin Li, Vikas Singh. Nyströmformer: A Nyström-based Algorithm for Approximating Self-Attention. AAAI 2021.
- Sinong Wang, Belinda Z. Li, Madian Khabsa, Han Fang, Hao Ma. Linformer: Self-Attention with Linear Complexity. CoRR abs/2006.04768 (2020).
- Marin Orsic, Sinisa Segvic. Efficient semantic segmentation with pyramidal fusion. Pattern Recognit. 110: 107611 (2021).
- Ping Chao, Chao-Yang Kao, Yu-Shan Ruan, Chien-Hsiang Huang, Youn-Long Lin. HarDNet: A Low Memory Traffic Network. ICCV 2019

---

> ### Author Response · Authors · 2022-08-02
> **Reply to Reviewer HQEz**
>
> Dear Reviewer HQEz,
> Thank you for the detailed review.  We will address your comments below.
> ### Weakness 1: GPU-Friendly attention shows incremental contribution.
> > Refer to the response of common question 1.
>
> ### Weakness 2: hybrid convolution-transformer models have been proposed before.
> > Refer to the response of common question 2.
>
> ### Weakness 3: the proposed attention performs only slightly better than baselines.
> > Refer to the response of common question 3.
>
> ### Weakness 4: missing configuration EA + CA.
> >  We replace GFA+CA with EA(r=0.25)+CA and got 32.1\% mIoU. It proves that GFA is more effective than EA even under the configuration of associating with cross-resolution attention, and GFA+CA is still the best choice for different combinations of various attentions.
>
> ### Weakness 5: large training footprint.
> >  We test the runtime CUDA memory of RTFormer with batch size of per gpu set as 3 and crop size set as 512x1024, the memory occupied on one GPU is shown as below:
> RTFormer-base: 10.22G, RTFormer-slim: 3.6G.
> We can see that the CUDA memory consumed by RTFormer is acceptable, and 16G V100 is enough for training RTFormer even with larger batch size.
>
> ### Weakness 6: incomplete related works.
> >  We are sorry about the omission of the references in related works, and we will add them in the revision version.
>
> ### Question 1: the name of the proposed network.
> >  It is a good suggestion, and we will take it into consideration.
>
> ### Question 2: showing experiments on some embedded platform.
> >  Due to the limited time and resources, we don't measure RTFormer on embedded platforms. We will show more experiments in the final version.
>
> ### Question 3: clarify whether you use V100 16GB or V100 32 GB.
> > We use V100 16G for training and testing.

---

### Official Review · Reviewer_dfdA · 2022-07-16

**Rating:** 5
**Confidence:** 5
**Soundness:** 3 good
**Presentation:** 2 fair
**Contribution:** 2 fair

**Summary:**

This paper studies the problem of real-time semantic segmentation with Transformer. The authors proposed an RTFormer block with two attention models to aggregate information on different-resolution features. The experimental results on serval datasets demonstrate the effectiveness of the proposed method.

**Questions:**

The authors argue that GPU-Friendly attention is quite friendly for GPU-like devices. However, according to Table 3, GPU-Friendly attention has a minor effect on acceleration. (189.6->189.8 or 187.9->189.8). Meanwhile, there is no fair and clear comparison between EA or other lightweight attentions and the proposed attention.


**Limitations:**

Yes

**Strengths And Weaknesses:**

[Strengths]
+ The proposed method achieves great performance in the serval datasets.
+ Compared to the baselines, the proposed methods could bring constant improvements

[Weaknesses] Some important ablation studies are missing.
- The choice of architectural design. The author put the proposed RTFormer block on the last two stages and does not provide the results to support this design.
- The baseline which does not use any attention is needed to be included in Table 3 (a).
- Comparison with the other lightweight attentions is also needed.

---

> ### Author Response · Authors · 2022-08-02
> **Reply to Reviewer dfdA**
>
> Dear Reviewer dfdA,
> Thank you for appreciating our approach. We will address your comments below.
> ### Weakness 1: The choice of architectural design.
> > We replace the attention blocks in stage3, stage4 and stage5 with GFA+GFA, and we got 31.5 mIoU. It is lower than the result 32.8, which configures GFA+GFA only in stage4 and stage5. This indicates that using attention in early stages is not quite efficient, and our proposed architecture design is relatively reasonable.
>
> ### Weakness 2: The baseline which does not use any attention.
> > We replace GFA with a sequential of two convolution layers on both high-resolution and low-resolution branches within the architecture of RTFormer, as shown in the Extention1 of Table 3a, we got 31.5 mIoU which is worse than using attention block
>
> ### Weakness 3: Comparison with the other lightweight attentions.
> > We compared GFA with two lightweight linear attentions, which are Linformer attention (LA)and Nyströmformer attention (NA). We replace the attention block with Linformer and Nyströmformer based on the experiment GFA+CA in Extention1 of Table 3a, and our method got the best performance and speed trade-off. Specifically, our proposed GFA+CA achieves same mIoU as the NA(64)+NA(64) while outperforms NA(64)+NA(64) significantly on speed by $160$\%. In contrast to Linformer attention, GFA+CA improves mIoU from $32.4$\% to $33.0$\% while maintaining $12$\% speed gain.
>
> ### Question1: GPU-Friendly attention has a minor effect on acceleration.
> > Refer to the response of common question 3.

---

### Author Response · Authors · 2022-08-02
**To all reviewers**

# To all reviewers
## Dear all reviewers:
We sincerely appreciate the reviewers for the time and efforts on the review. We first explain the common questions about the novelty of GPU-Friendly attention and the hybrid structure of RTFormer, followed by detailed responses to each reviewer separately. In addition, we provide more ablation study results mentioned by reviewers in the below two tables.

Extention1 of Table 3a.  Comparison among different types of attention on ADE20K. SA, EA, GFA, CA, LA, NA denote multi-head Self-Attention, multi-head External Attention, GPU-Friendly Attention, Cross-resolution Attention, Linformer Attention and Nyströmformer Attention respectively. For example, GFA+CA means adopting GFA in low-resolution branch and CA in high-resolution branch. r is a ratio for adjusting the parameter dimension $M$ in multi-head external attention. C is the number of base feature dimension in network (default C is 32). NA(32), NA(64) denote the Nystromformer Attention with 32 landmark points and the Nystromformer Attention with 64 landmark points respectively.
| Attention       | GPU        | FPS of whole network ↑ | val mIoU(%) ↑ |
|-----------------|------------|:----:|:----:|
| Conv+Conv       | RTX 2080Ti |         191.3              | 31.5          |
| SA+SA           | RTX 2080Ti |        97.4            | 32.7          |
| EA+EA (r=0.125) | RTX 2080Ti |       196.9            | 31.9          |
| EA+EA (r=0.25)  | RTX 2080Ti |       189.6            | 32.0          |
| EA(r=0.25)+CA   | RTX 2080Ti |       189.1            | 32.1          |
| EA+EA(r=1)      | RTX 2080Ti |       180.8            | 32.2          |
| EA+EA(r=1,C=36) | RTX 2080Ti |       134.8            | 32.8          |
| LA+LA           | RTX 2080Ti |       167.6            | 32.4          |
| NA(32)+NA(32)   | RTX 2080Ti |        77.6            | 32.9          |
| NA(64)+NA(64)   | RTX 2080Ti |        72.2            | 33.0          |
| GFA+GFA         | RTX 2080Ti |       189.8            | 32.8          |
| GFA+CA          | RTX 2080Ti |       187.9            | 33.0          |


Extention2 of Table 3a: latency of different attention blocks. We use PyTorch Profiler to analyze latency of different attention blocks. To achieve stable comparison, we use $100$ times to warm up devices and measure the blocks with  $100$ times on RTX 2080Ti GPU with the same input shape(1, 64, 64, 256). The CUDA time total is obstaned by PyTorch Profiler key\_averages() API.
NA(64), NA(32), EA, LA, GFA denote Nyströmformer Attention with 64 landmark points, the Nyströmformer Attention with 32 landmark points, External Attention, LinFormer Attention, and GPU-Friendly Attention respectively.

| Attention       | GPU        |  CUDA time total↓|
|-----------------|------------|:----:|
| NA(64)          | RTX 2080Ti |        304.308ms   |
| NA(32)          | RTX 2080Ti |         277.195ms  |
| LA              | RTX 2080Ti |          16.081ms      |
| EA(r=1)         | RTX 2080Ti |         19.088ms     |
| GFA             | RTX 2080Ti |         10.993ms      |

---

> ### Author Response · Authors · 2022-08-02
> **A.1-1 Common Question 1: Novelty of GPU-Friendly Attention**
>
> * Difference with External Attention
>
> >  Although GFA is derived from EA, but it has essential difference with EA. The major contribution of GFA over EA is to make attention operation achieving better trade-off on GPU-like devices. Multi-head mechanism can enlarge the capacity of network as it calculates multiple attention maps during token mixing. But this makes the matrix multiplication be batch-wise which is not inference friendly on GPU like devices. Thus, increasing the computation efficiency of attention on GPU while maintaining the capacity introduced by multi-head mechanism is the breakthrough of GFA.
>
> > In order to be inference friendly on GPU, two main principles should be considered. One is that the matrix multiplication should be integrated instead of batch-wise. Another is the number of operations in sequential should be small as launch CUDA kernel is time cost. However, multi-head EA can't satisfy these two principles. Firstly, batch-wise matrix multiplication is needed due to the multi-head mechanism. And it becomes slow more seriously than vanilla matrix multiplication when the dimension of feature increases. Secondly, different heads do not communicate when calculating token mixing, thus additional projection layers are needed for fusing channel information among different heads.
>
> > In consideration of the above problems, our GPU-Friendly attention does not split the feature into multiple heads, while uses vanilla matrix multiplication to compute token mixing directly instead of conducting batch-wise matrix multiplication, and uses grouped normalization to enhance the ability of GPU-Friendly attention to capture distinct information from different groups of feature. Furthermore, as vanilla matrix multiplication already has the capability of fusing channel information, the additional projection layer is not needed any more. As a result, GPU-Friendly attention simplifies the multi-head EA from 5 operations (two projection layers, two batch-wise matrix multiplications for token mixing and an activation) to 3 operations (two vanilla matrix multiplications for token mixing and an activation), which is more suitable for GPU inference.
>
> > While, the reason that GPU-Friendly attention can maintain the capacity introduced by multi-head mechanism is the introduction of grouped normalization. In order to make it easier to be comprehended, we can hypothesis a special case that the external parameters $K_g$ and $V_g$ of GFA are both block diagonal matrices, and the number of blocks is equal with the number of groups in grouped normalization. Under this condition, GFA degrades to a variation of multi-head EA. Features are spilted into several heads, and each head of feature is calculated with the corresponding parameter block to generate attention map. So GFA has the fully ability of multi-head mechanism to capture distinct information from different parts of feature. Furthermore, this indicates that GFA using dense external parameters is more generalized than multi-head EA, as it calculates attention maps from all channels of token feature by the assistance of grouped normalization, and distinct information are extracted softly by the dense external parameters.
>
> > Although the final formulation of GPU-Friendly is simple, but it extends the capability of multi-head EA. And just because the simplicity of GFA, more efficient inference on GPU-like devices is achieved.

---

> > ### Author Response · Authors · 2022-08-02
> > **A.1-2 Common Question 1: Novelty of GPU-Friendly Attention**
> >
> > * Difference with Linformer
> > > The Linformer reduces the complexity to O(n) through projecting the length dimension of keys and values to a lower dimensional space.
> > Compared with the Linformer, GPU-Friendly attention has two differences.
> > Firstly, similar to EA, GFA takes the keys and values as the learnable parameters. Thus, there is no need to have extra projection matrices to generate keys and values.
> > Secondly, the attention in Linformer is still built upon on multi-head mechanism which is slower than our GPU-Friendly attention shown in the Extension2 of Table 3a . And the reasons why multi-head mechanism is slow has been given in difference with External Attention part.
> > Also, the performance and FPS comparison between the attention of the Linformer and RTFormer are shown in Extersion1 of Table 3a, which can further demonstrate the difference between the Linformer attention and our GFA.
> >
> > * Difference with Nyströmformer
> > > The Nyströmformer repurposes the Nyström method for approximating self-attention computation. The Nyströmformer achieves linear computation by adopting landmarks points to reconstruct the softmax matrix. Although the Nyströmformer reduces the entire computational cost of self-attention, it splits the original softmax matrix computation into several parts and then combines them. Splitting the big matrix computation into different parts will cause the suboptimal inference efficiency on GPU-like devices. Besides of this splitting operation, Nyströmformer also has the problems brought by the vanilla multi-head mechanism which has already been explained in Difference with External Attention part.
> > According to Extension2 of Table 3a, our GPU-Friendly Attention can achieve the lowest latency on GPU among these different types of attention. Also, the RTFormer can achiever better performance and speed trade-off than Nystromformer.

---

> > > ### Author Response · Authors · 2022-08-02
> > > **A.2 Common Question 2: Novelty of hybrid structure**
> > >
> > > > Indeed, hybrid structure for semantic segmentation task is not new, such as DPT-hybrid, HRNet+OCR, HRFormer. But they are mostly non real-time architectures. To our best known, RTFormer block is the first one to apply transformer attention to real-time semantic segmentation task. From the perspective of micro structure, we propose a simple GPU-Friendly attention which can be simply constructed by vanilla matrix multiplication. Beside, we also propose to use 3x3 convolution in FFN and apply cross-resolution attention to construct network module. Within RTFormer block, data flows across all matrix operations are intuitively more friendly to GPU devices than other attention modules which contain relatively more time-consuming operations. Surprisingly, because of the introduction of RTFormer block, RTFormer becomes more general than other structures for real-time semantic segmentation task due to the constant improvements over Cityscapes, CamVid, ADE20K and COCO-Stuff datasets. Most importantly, this hybrid structure solves a critical issue that how to make full use of transformer in real-time segmentation scenario, and it provides a perspective for further research to excavate the usage of transformer in real-time scenario more deeply.

---

> > > > ### Author Response · Authors · 2022-08-02
> > > > **A.3 Common Question 3: About the improvement of performance and efficiency in Table 3a**
> > > >
> > > > ### A.3 Common Question 3: About the improvement of performance and efficiency in Table 3a
> > > > > First of all, it is worth to be noted that the performance and efficiency are both important in the comparison of different methods. Secondary, the comparison of FPS in Table 3a is about the whole network. For example, GFA+CA achieves 187.9FPS which is aligned with the FPS of RTFormer-Slim in Table 2 from our paper. We are sorry that this setting is not clearly clarified in Section4.4 and we will fix it in the revision version. In order to show the improvement of GFA over other existing attentions clearly, we supplement several experiments and evaluations and then elaborate the issue from two aspects.
> > > >
> > > > > As shown in Extention2 of Table 3a, from the view of pure attention block, we report the latency of different types of attention block on the high resolution branch of stage4. As shown in Extention2 of Table 3a, about 11ms is consumed by GFA while 19ms for EA. This illustrates that our GFA outperforms EA greatly on CUDA speed by 73\%. Meanwhile, we also report the latency of other efficient attentions like linformer attention and Nyströmformer attention, and all of them are also slower than GFA.
> > > >
> > > > > As shown in Extention1 of Table 3a, from the view of the whole network, we report the FPS of different networks. As shown from the comparison between EA+EA(r=1) and GFA+GFA in Extention1 of Table 3a, GFA outperforms EA on both efficiency and performance, and it is not easy in real-time segmentation task. Further more, we take an experiment that enlarging the base feature dimension of EA+EA(r=1) from C=32 to C=36, then we got 32.8 mIoU, while the FPS drops from 180.8 to 134.8. From this result, we can conclude that the network equipped with GFA+GFA is faster than EA+EA about 41\% when they achieve the same performance, and this improvement is considerable.

---

### Author Response · Authors · 2022-08-09
**about the review**

Dear reviewers,
We appreciate all the reviewers for the time and effort on the review.
We have added some experiments and answered questions.  Because the discussion time is coming to an end, we sincerely hope to communicate with the reviewers again.
Looking forward to your reply.

---

### Author Response · Authors · 2022-08-09
**To all reviewers**

Dear reviewers,
Since the rebuttal discussion is about to end soon, it would be better to let us know whether our replies have addressed you questions.
And don't hesitate to contact us if you have any further clarifications required.

---

### Meta-Review · Area_Chair_kfJD · 2022-08-27

**Recommendation:** Accept
**Confidence:** Certain

**Metareview:**

Reviewers agree that the proposed RTFormer block and overall network architecture achieves good trade-off between performance and efficiency on several datasets. The design of GPU-friendly attention and cross-resolution attention improves the computational efficiency over multi-head attention, and well captures global context information when updating high-resolution embeddings.

The main concern, as mentioned by several reviewers, is the overall novelty as some ideas are related to previous work (GPU-friendly attention and hybrid convolutional-transformer architecture). Other issue includes missing baselines that are based on light-weighted attention designs, or do not use attention at all, but this have been well resolved in the author feedback. In summary, the pros outweigh the cons and therefore AC recommends acceptance.

**Award:**

No

---

### Decision · Program_Chairs · 2022-09-14

Accept